

# Contribution of Potential Evaporation Forecasts to 10-day streamflow forecast skill for the Rhine river

Bart van Osnabrugge[1,2], Remko Uijlenhoet[2], and Albrecht Weerts[1,2]

[1]Deltares, Operational Water Management Department, Delft, The Netherlands
[2]Wageningen University, Hydrology and Quantitative Water Management Group, Wageningen, The Netherlands

**Correspondence:** Bart van Osnabrugge (Bart.vanOsnabrugge@deltares.nl)

**Abstract.** Medium term hydrologic forecast uncertainty is strongly dependent on the forecast quality of meteorological variables. Of these variables, the influence of precipitation has been studied most widely, while temperature, radiative forcing and their derived product potential evapotranspiration (PET) have received little attention from the perspective of hydrological forecasting. This study aims to fill this gap by assessing the usability of potential evaporation forecasts for 10-day-ahead streamflow forecasting in the Rhine basin, Europe. In addition, the forecasts of the meteorological variables are compared with observations.

Streamflow reforecasts were performed with the daily wflow_hbv model used in previous studies of the Rhine using the ECMWF 20-year meteorological reforecast dataset. Meteorological forecasts were compared with observed rainfall, temperature, global radiation and potential evaporation for 148 subbasins. Secondly, the effect of using PET climatology versus using observation-based estimates of PET was assessed for hydrological state and for streamflow forecast skill.

We find that: (1) there is considerable skill in the ECMWF reforecasts to predict PET for all seasons, (2) using dynamical PET forcing based on observed temperature and satellite global radiation estimates results in lower evaporation and wetter initial states, but (3) the effect on forecasted 10-day streamflow is limited. Implications of this finding are that it is reasonable to use meteorological forecasts to forecast potential evaporation and use this is in medium-range streamflow forecasts. However, it can be concluded that an approach using PET climatology is also sufficient, most probably not only for the application shown here, but for most models similar to the HBV concept and for moderate climate zones.

As a by-product, this research resulted in gridded datasets for temperature, radiation and potential evaporation based on the Makkink equation for the Rhine basin. The datasets have a spatial resolution of 1.2x1.2 km and an hourly timestep for the period from July 1996 through 2015. This dataset complements an earlier precipitation dataset for the same area, period and resolution.

## 1 Introduction

Hydrologic forecasting has the aim to predict the future state of important hydrologic fluxes, most notably streamflow. Throughout the process of forecasting, from model set-up via initial state estimation to forecast run, meteorological forcing is a key component. Precipitation is known to be the main driver of hydrological processes and most of the forecast uncertainty is





attributed to inaccurate precipitation forcing (Cuo et al., 2011; Pappenberger et al., 2005). As a consequence, most attention has been given to the accuracy of precipitation forecasts. See for example the review of Cloke and Pappenberger (2009).

Evaporation is a result of the interaction between meteorological forcing and, physical and physiological, processes at the land surface. Meteorological forcing provides the potential energy (potential evaporation or PET) for evaporative processes
to take place. There are many formulas to estimate the potential energy available for evaporation, which can be divided in three types of formulas based on their data requirements (Xystrakis and Matzarakis, 2011; Xu and Singh, 2002): Temperature-based (e.g. Hargreave equation, Hammon's equation), radiation-based, and combined methods (e.g. Hansen's equation, Turc's equation, Makkink's equation). From an operational viewpoint the different types of formulas result in different demands on data availability.

Constraints on data availability have led to additional approximations for potential evaporation. A common approximation is the calculation of a monthly potential evaporation climatology or PET demand curves [Anderson, 2002]. This climatology is then used as driver for both historic potential evaporation and future potential evaporation.

Hydrological models have proven to be insensitive to the difference between variable potential evaporation forcing and monthly potential evaporation forcing with respect to the model's potential to estimate streamflow after calibration (Andréas-
sian et al., 2004; Oudin et al., 2005a, b). However, in forecasting, different choices in the handling of forcing data can be made between the historic update step and the forecast step, while the hydrological model, as a rule, remains the same. It therefore remains relevant to understand how a single model reacts to potential evaporation forcing. Insensitivity to the type of potential evaporation during the process of calibration does not mean that a model is insensitive to the form of potential evaporation input.

As mentioned above, there has been little attention of the forecast skill of the secondary forcing variables temperature and radiation in the hydrological context of potential evaporation. Furthermore, there is an easy and often used practice of avoiding potential evaporation forecasts by using a potential evaporation climatology. Therefore, the objective of this study is to assess to what extent potential evaporation forecasts can contribute to streamflow forecast skill.

This question is evaluated for the Rhine basin in Europe (fig 1). The Rhine is one of the basins currently employed as case
study for the IMproving PRedictions of EXtremes (IMPREX) project, which aims to improve predictions and management of hydrological extremes through climate services (van den Hurk et al., 2016).

Several studies already directly addressed some aspects of operational ensemble flow forecasts in the Rhine. Renner et al. (2009) showed that at the time meaningful hydrological ensemble forecasts could be produced up to a 9 day leadtime for the Rhine river based on ECMWF ensemble meteorological forecasts. Reggiani et al. (2009) used a Bayesian ensemble uncertainty
processor to improve the assessment of uncertainty in the ensemble forecast. Terink et al. (2010) applied downscaling techniques to ERA15 ECMWF reanalysis data to develop a downscaling strategy for regional climate models (RCMs). Verkade et al. (2013) developed post-processing techniques to improve the precipitation and temperature ECMWF forecasts for the hydrological model. Photiadou et al. (2011) compared two historical precipitation datasets and assessed the influence of precipitation datasets on model results. Recently, van Osnabrugge et al. (2017) developed a high resolution hourly precipitation
dataset for use with gridded hydrologic models.





To answer the research question model experiments are performed, but first the data and hydrological model are presented (Section 2). Second, the model experiments are described, which also partitions the main question into three subquestions (Section 3) which are subsequently answered (Section 4). The paper concludes with a discussion on the results in the wider context of evaporation modelling in hydrologic forecasting and the conclusions (Section 5).

## 2 Data and Model

Observational data has been preprocessed for use with a grid based hydrological model. The data was processed with hourly time resolution, on a 1.2x1.2 km grid spatial resolution, and for the period mid 1996 through 2015. All source data to derive the gridded estimates comes from sources that supply their data in near real-time making the datasets suitable for operational forecasting. For this study all data was aggregated to a daily time step. The hourly datasets are downloadable through the 4TU data centre (van Osnabrugge, 2017, 2018).

### 2.1 Precipitation

For this study the precipitation dataset is used which has been derived by van Osnabrugge (2017). The precipitation is derived using the genRE interpolation method based on ground measurements and the HYRAS (Rauthe et al., 2013) climatological precipitation dataset (van Osnabrugge et al., 2017).

### 2.2 Temperature

Temperature observations (1996-2016) are interpolated on the same 1.2x1.2 km grid as the precipitation data. Temperature is derived from interpolation of ground measurements with correction for height using the SRTM digital elevation model (Farr et al., 2007) and standard lapse rate as follows.

To calculate temperature $T_x$ at a given grid cell x from a number of n surrounding stations, determine a set of weights based on inverse-distance squared weighting between all stations (typically the n closest stations) and the grid cell. This step can have a threshold for maximum distance. $d_{i,x}$ is the distance between station $i$ and cell $x$:

$$w_{i,x} = \frac{1/d_{i,x}^2}{\sum_{i=1}^{n} 1/d_{i,x}^2} \tag{1}$$

Second, interpolate the measured temperature $T_{m,i}$ with the weights as with standard inverse-distance squared interpolation:

$$T_{m,x} = \sum_{i=1}^{n} T_{m,i} w_{i,x} \tag{2}$$

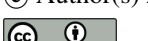



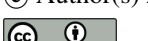

**Figure 1.** Map of the Rhine basin, Europe. Black lines delineate 148 subbasins used in the analysis of the meteorological forecast skill.
Square markers show the locations used for forecast skill analysis.   **4**





Third, calculate the temperature lapse correction term $T_{\gamma,x}$ as the weighted difference between the height of the grid cell $H_x$ and the height of the considered stations $H_i$ multiplied with the lapse rate $\gamma$.

$$T_{\gamma,x} = \gamma \left( \sum_{i=1}^{n} (H_i - H_x) w_{i,x} \right) \qquad (3)$$

Note that $T_{\gamma,x}$ is static for a fixed configuration of the measurement network if $\gamma$ is taken to be a constant. In this study the

configuration of the measurement network changed based on the number of reporting stations at each time step. A constant lapse rate was assumed: $\gamma = 0.0066 [^\circ C/m]$.

The final temperature estimate for grid cell $x$ is obtained by adding $T_{\gamma,x}$ and $T_{m,x}$:

$$T_x = T_{\gamma,x} + T_{m,x} \qquad (4)$$

## 2.3 Downwards Shortwave Surface Radiation Flux

The availability of solar radiation measurements at the surface has proven to be spatially and temporally inadequate for many applications, with remotely sensed solar radiation products having the largest potential to remedy this (Journée and Bertrand, 2010). Remotely sensed solar radiation estimates from the Land Surface Analysis Satellite Application Facility (LSA-SAF) were found to be in closer agreement with ground observations than reanalysis datasets such as the Système d'Analyze Four-nissant des Renseignements Atmosphériques à la Neige (SAFRAN) reanalysis (Carrer et al., 2012) and ERA-Interim (Jedrzej

et al., 2014).

For this study, downward shortwave radiation is resampled and merged from the EUMETSAT Surface Incoming Solar Radiation (SIS) (Mueller et al., 2009) and Downward Surface Shortwave Flux (DSSF) (Trigo et al., 2011) products from the Climate Monitoring Satellite Application Facility (CM-SAF) and LSA-SAF, respectively. Gaps in the satellite data are filled with the ERA5 surface solar radiation downwards (ssrd) parameter from the 4d-var reanalysis (Copernicus Climate Change

Service, 2018). ERA5 was found to have comparable mean bias with satellite-derived products for inland stations (Urraca et al., 2018).

In earlier research it has been shown that LSA-SAF (2005-current) and CM-SAF (1983-2005) can consistently be merged into one time series (Jedrzej et al., 2014). The products of the different SAFs are comparable in terms of bias and standard deviation (Ineichen et al., 2009).

## 2.4 Makkink Potential Evaporation

There are different approaches in making use of remotely sensed data to calculate evapo(transpi)ration. One branch of research aims to calculate actual evapotranspiration directly from satellite imagery (Su, 2002). Applications range from estimating the global evaporation flux (Mu et al., 2011), water resources management (Bastiaanssen et al., 2005) and constraining model parameters for a gridded model (Immerzeel and Droogers, 2008).

For operational use, PET estimates can be derived from satellite data only, or from a combination of satellite imagery and ground measurements. Bowman et al. (2017) explored the use of MODIS to provide a daily PET, both as dynamic PET (Spies



et al., 2015) and PET climatology (Bowman et al., 2016) for a gridded and lumped version of the Sacramento Soil Moisture Accounting (SAC-SMA) model. The model was recalibrated for each PET input. No configuration with MODIS derived PET showed consistent improvements across all basins in their case study. Still, it was concluded that the combination of dynamic PET in combination with a gridded model had the best overall results (Bowman et al., 2017).

A disadvantage of using satellites such as MODIS is their temporal coverage which is restricted to a single overpass at a set time each day giving one instantaneous value. This can be resolved by assuming a sinusoidal development of PET over the day (Kim and Hogue, 2008), but the limitation is clear. This disadvantage is resolved by using geostationary satellites. For example, Jacobs et al. (2009) used solar radiation from NOAA GOES geostationary satellite in combination with ground observations to calculate daily PET with the Penmann-Montheith equation.

Here, potential evaporation is calculated from geostationairy satelite radiation estimates and ground observations of temperature with the method proposed by Makkink, which is applicable with remote sensed radiation estimates (de Bruin et al., 2016). PET calculated with Makkink's equation is a reference crop evapotranspiration, which means that crop factors apply determined by the hydrological model. In the set-up of our hydrological model the crop factor was determined by land-use. A crop factor of 1.15 is applied to the forested areas and 1.0 to all others.

The reasons for choosing the Makkink equation are that 1) it only needs radiation and temperature, for which gridded estimations are available and 2) the Makkink equation is used by the Royal Netherlands Meteorological Institute (KNMI) so that the work presented here is compatible with ongoing local research (Hiemstra and Sluiter, 2011).

The potential evaporation is calculated based on air temperature T [$°C$] and downward shortwave radiation $R_g$ [$Wm^{-2}$] for accumulation period $t$ [$s$] (Hiemstra and Sluiter, 2011):

$$PET = 1000 \cdot 0.65 \frac{\Delta}{\Delta + \psi} \cdot \frac{tR_g}{\lambda \rho_w} [mm] \tag{5}$$

with, $\psi$ the psychrometric constant, $lambda$ the latent heat of water, $\Delta$ the slope of the saturation vapor pressure curve and $\rho_w$ the density of water calculated by:

$$\psi = 0.646 + 0.0006T [hPa°C^{-1}] \tag{6}$$

$$\lambda = 1000(2501 - 2.38T)[Jkg^{-1}] \tag{7}$$

$$\Delta = \frac{6.107 \cdot 7.5 \cdot 273.3}{(273.3 + T)^2} e^{\frac{7.5T}{273.3+T}} [hPa°C^{-1}] \tag{8}$$

$$\rho_w = 1000[kgm^{-3}] \tag{9}$$

The Makkink Potential Evaporation calculated for each time step is called 'near real-time' ($PET_{NRT}$). The potential evaporation climatology ($PET_{Clim}$) was calculated by averaging over the full time period (20yr) for each day (fig. 2).



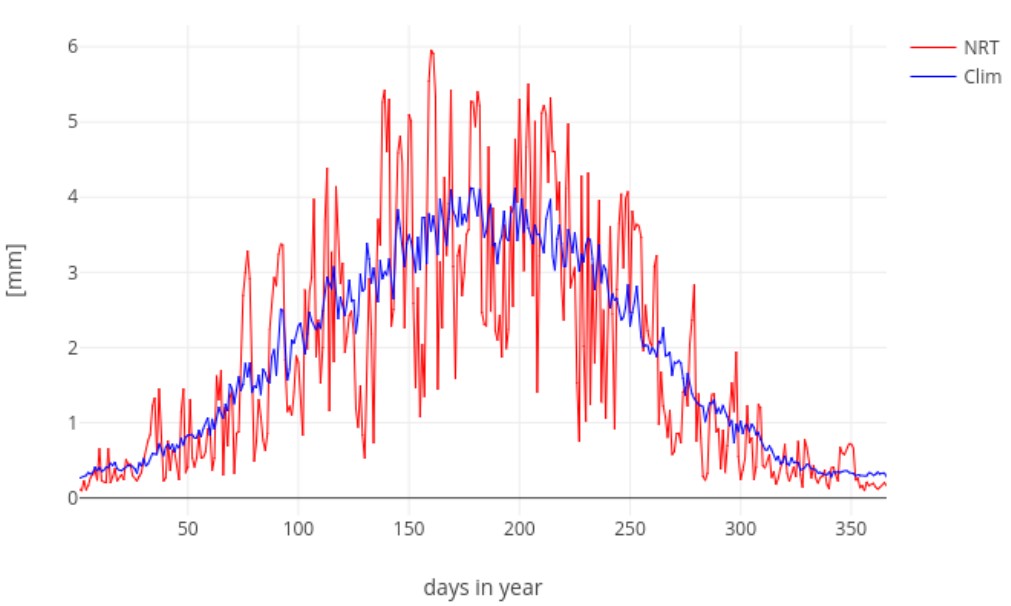

**Figure 2.** Difference between climatology and near real time potential evaporation. Shown for the year 2004 for grid-cell x:200, y:200.

## 2.5 ECMWF reforecast

The European Center for Medium-Range Weather Forecasts (ECMWF) issues hindcasts produced with the current model cycle for certain days for the last 20 years. The reforecast obtained for this study was produced with model cycle 43r1 (Buizza et al., 2017). The first forecast is on 1996-03-10 and the last forecast on 2015-12-29 with reforecasts alternating every three or four days.

Forecasted Makkink potential evaporation ($PET_{Fcast}$) is calculated based on the $t2m$ ($T$) and $ssrd$ ($R_g$) variables using equations 5-9. Temperature was first downscaled to the model resolution using the standard lapse rate as used in the interpolation of the temperature observations as follows:

$$T_x = \overline{T} + \left(\overline{h} - h_x\right)\gamma \tag{10}$$

With, $\overline{T}$ the temperature given by the ECMWF forecast on the ECMWF resolution, $\overline{h}$ the average height of the DEM corresponding to the footprint of the ECMWF grid cell, $h_x$ the height of cell x in the model, and $\gamma$ the lapse rate.



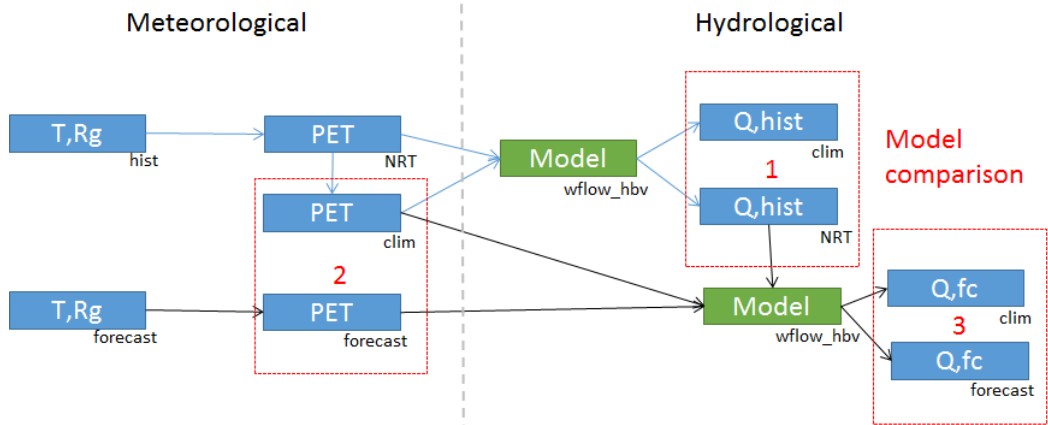

**Figure 3.** Flow chart of the model experiment. Blue boxes represent data products. Green boxes depict modeling activities. Arrows represent the flow of data for historical runs (blue lines) and forecast runs (black). The red boxes indicate the areas for analysis of the results, each box targeting a research subquestion.

## 2.6 Hydrological Model

wflow is a modular hydrological modelling framework that allows for easy implementation and prototyping of regular grid hydrological model concepts in python-pcraster (Schellekens et al., 2017). The hydrological model concept used is the HBV (Hydrologiska Byråns Vattenbalansavdelning) model concept (Lindström et al., 1997) applied on a grid basis. The generated

5    runoff is routed through the river network with a kinematic wave approach (Schellekens et al., 2017). In the following this model is referred to as wflow_hbv. The set-up of the hydrological model is the same as used in assessing the validity of the genRE precipitation data set (van Osnabrugge et al., 2017). The model was parameterized through calibration with a Generalised Likelihood Uncertainty Estimation (GLUE) like procedure (Beven and Binley, 1992), using HYRAS precipitation as forcing data (Winsemius et al., 2013, ?). The model is taken 'as is' and is not recalibrated for each PET forcing, the effect of which has

10   been studied extensively elsewhere (e.g. Bowman et al., 2017; Oudin et al., 2005a).

## 3   Experimental Set-up

The analysis consists of a meteorological part and an hydrological part (fig. 3).

### 3.1   Analysis of meteorological forecast skill

In this analysis we aim to answer the following question:

15      1. What is the forecast skill of temperature, radiation and potential evaporation compared to precipitation?



For this purpose the observations and forecasts are spatially averaged over 148 subbasins fig. (1). Time series of observations and forecasts are then used to calculate the Mean Continuous Ranked Probability Skill Score (CRPSS) for each basin and each season (MAM,JJA,SON,DJF).

The Mean Continuous Ranked Probability Score (CRPS) is an overall measure of forecast quality and is calculated by:

$$\overline{CRPS} = \frac{1}{n}\sum_{i=1}^{n}\int_{-\infty}^{\infty}\left(F_y(y) - \mathbf{H}(y \geq x)\right)\mathrm{d}y \tag{11}$$

In which $F_y(y)$ is the cumulative distribution function of the forecast variable and $\mathbf{H}(y \geq x)$ the Heaviside step function that assumes probability 1 for values greater than or equal to the observation and 0 otherwise (Brown et al., 2010). Interpretation of the mean CRPS is similar to interpretation of a Root Mean Square Error. Both scores have no fixed limit, their magnitude is determined by the variable, and lower scores are better with zero the perfect score.

The limits of the mean CRPS vary depending on the basin and season and it is therefore difficult to compare between basins and season. For this reason the CRPS is translated into the Continuous Ranked Probability Skill Score, which measures the performance of a forecasting system relative to a reference forecast. The reference forecast here is seasonal climatology. As such the CRPSS equals 1 for a perfect forecast and 0 when the forecast ensemble does not score a better CRPS than the CRPS calculated for the climatological distribution.

$$\overline{CRPSS} = \frac{\overline{CRPS_{REF}} - \overline{CRPS}}{\overline{CRPS_{REF}}} \tag{12}$$

Additionally the Relative Mean Error (RME) is calculated for the mean of the forecasts $\overline{Y_i}$ to detect relative biases in the mean:

$$RME = \frac{\sum_{i=1}^{n}(\overline{Y_i} - x_i)}{\sum_{i=1}^{n}x_i} \tag{13}$$

In which $\overline{Y_i}$ is the mean of the ensemble for forecast $i$ and $x_i$ the corresponding observation.

The above scores are calculated with the Ensemble Verification System (EVS), a software package to calculate ensemble verification metrics (Brown et al., 2010).

### 3.2 Analysis of the effect of PET forecasts on streamflow predictions

In this second part of the analysis we aim to answer the following questions:

1. To what extent are initial states affected by the use of climatological versus near real time potential evaporation?

2. To what extent can potential evaporation forecasts contribute to streamflow forecast skill?





To answer the first question, the wflow_hbv model is consecutively forced with $PET_{Clim}$ and $PET_{NRT}$. Four states and two fluxes are exported for analysis: 1) upper soil reservoir, 2) lower soil reservoir, 3) interception storage, 4) soil moisture store; and fluxes 5) discharge and 6) actual evaporation. For the different states and fluxes the Mean Difference (MD) is calculated for each grid cell. This is done for each season to investigate seasonality of differences. The MD is calculated as:

$$MD = \frac{\sum_{i=1}^{n}(STATE_{NRT,i} - STATE_{CLIM,i})}{n} \tag{14}$$

To answer the second question two hindcast runs are performed with $PET_{Fcast}$ and $PET_{Clim}$ as PET forcing, respectively. To avoid effects caused by the initial state all forecasts start from the initial states derived from the $PET_{NRT}$ simulation. Forecast skill scores are calculated as for the meteorological variables for 20 discharge gauges and for each season. Different from the meteorological verification exercise, the metrics are calculated for the forecasts with reference to the model output and not compared with observations. The reason for this was that differences between observation and forecast stem from many different sources, including errors in the initial state. Subsequently, a forecast that is 'too wet' might compensate in the 10 day forecast for initial states that were 'too dry'. For this reason the effect of the meteorological forecast was isolated by calculating the verification metrics against modeled streamflow. This also avoids issues of perceptive bias due to the model being calibrated on another PET forcing; One of the PET types might simply perform better because it is more like the original PET used in calibration.

Streamflow gauges for analysis were selected such that:

1. Only gauges were chosen for which the model was deemed behavioral as expressed by a KGE score threshold of 0.5.

2. Only one gauge was selected for each stream in the basin, except for the Rhine river itself, for which 2 additional gauges were chosen. If multiple gauges in the same stream were present the gauge most downstream was chosen. The gauge 'most downstream' was selected by sorting on mean yearly discharge and picking the highest.

3. From the then remaining list, the largest 20 streams were selected for analysis.

The streamflow locations are shown in figure 1 as black squares including the name of the river.

## 4    Results

### 4.1    Analysis of meteorological forecast skill

The forecast skill is assessed for all catchments and for each season. Seasons are northern hemisphere seasons spring (MAM), summer (JJA), autumn (SON), and winter (DJF). Figure 4 shows the Mean Continuous Ranked Probability Skill Score (CRPSS) calculated for subsamples of all forecast-observation pairs for different levels of over exceedance, $P(x \geq X)$, for each variable. Simply put, the CRPSS value at $P(x \geq X) = 0.9$ is calculated for the top 10% of observations and the CRPSS value at $P(x \geq X) = 0.1$ is calculated for the highest 90% of the observations. $P(x \geq X)$ is calculated over all observations from all





**Figure 4.** Continuous Ranked Probability Skill Score (CRPSS) for the four forcing variables benchmarked against sample climatology for the 148 HBV subbasins. CRPSS scores are aggregated into mean (solid), 10th and 90th percentile (dashed).

seasons. This means that for some seasons, for example temperature in winter, there is an upper limit in $P(x \geq X)$, because the highest temperatures do not occur during winter. On the other hand, the response of the CRPSS curve is flat for low $P(x \geq X)$ for temperature during summer as all summer temperatures fall in the highest 60% of temperatures of the whole year. The same is shown for the Continuous Ranked Probability Score (CRPS), fig. 5, and the Relative Mean Error (RME), fig. 6.

5     There is no skill in the ECMWF forecast beyond 10 days for daily precipitation. This is consistent with the 9-day leadtime in streamflow forecasts found by Renner et al. (2009). The skill is best in winter and worst in summer, which is expected based on the dominating meteorological processes (frontal systems in winter and convective events in summer). The total amount of precipitation is underestimated after one-day lead time (fig. 6).





**Figure 5.** Continuous Ranked Probability Score (CRPS) for the four forcing variables benchmarked against sample climatology for the 148 HBV subbasins for the whole year. CRPS is aggregated into mean (solid), 10th and 90th percentile (dashed).



**Figure 6.** Relative Mean Error (RME) for the four forcing variables benchmarked against sample climatology for the 148 HBV subbasins for the whole year. RME is aggregated into mean (solid), 10th and 90th percentile (dashed).





There is more skill in the forecast for the variables temperature and incoming shortwave radiation. Likewise, there is considerable skill remaining in the potential evaporation forecast. For temperature the one-day forecast is close to perfect for autumn and spring. The skill in temperature forecast is similar for spring, summer and autumn, but worse during winter. The spread, the difference in skill between basins, is also largest during winter and spring. The RME shows that there is a small negative bias in the temperature forecasts. The RME for winter is largests, however it should be noted that the RME is the mean difference weighed by the mean of the observations (eq. 13). As the mean temperature in winter is closer to zero, this results in larger RME. Still, also when expressed in absolute values, the error for temperature during winter is larger than for other seasons (fig. 5).

For radiation there is already quite a considerable loss in skill after one day, but then the CRPSS remains quite stable for longer forecasts, notably during spring and autumn. There is a larger decline in skill for summer and for extreme low radiation values in winter. In absolute terms, the CRPS is related to the magnitude of the average radiation for each season, with the smallest absolute errors for winter and the largest during summer (fig. 5). In terms of bias, low values are slightly overestimated while high values are slightly underestimated, making the radiation forecasts slightly less extreme than the observations (fig. 6).

The skill of the potential evaporation forecast is closely tied to the skill in radiation forecast, both because Makkink potential evaporation is directly proportional to radiation and because the larger uncertainty in the radiation forecast. The forecast skill has the same properties as those found for the radiation forecast. A small difference is that some of the longer lead time found for temperature is found back in slighly improved forecast skill after 10 days for PET compared to radiation.

Overall, there is relatively little spread in skill between basins, with the 10th and 90th percentile close to the mean and following the same trajectory. The difference in skill between the different seasons is larger than the spread between basins, especially for the variables temperature, radiation and potential evaporation. This difference in skill between seasons is partly misleading. For example, the forecast skill for radiation in winter (fig. 4, purple line) appears to be an outlier. However, the whole range of occurrences of extreme high and low radiative forcing is compressed in a limited part of $P(x \geq X)$. Although the forecast over the whole range of winter radiative forcing is lower than that for the other seasons, the top 10% of winter radiative forcings are actually among the best predicted.

Likewise, high temperatures receive higher skill scores than low season temperatures. This is even more distinct in the radiation forecasts. This does, however, not mean that the forecasts of such rare events are more accurate: both RME (fig. 6) and CRPS (fig. 5) are larger for high extremes, meaning larger errors for those forecasts. Still, taking into account the rarity of the event by calculating the CRPSS, which is the skill of the forecast relative to the skill of a random draw from the climatology, temperature, radiation and potential evaporation forecasts are found to add most information for extreme high values, even though the error of those forecasts is larger than for more 'average' values (values with higher probability of occurrence).




## 4.2 Influence of dynamic PET on initial states

Dynamic potential evaporation leads to lower actual evaporation (AET). The difference is largest for summer and spring (fig. 7). Part of this lower evaporation is from a reduction in interception as the interception storage is more filled on average under dynamical forcing. This can be explained by the correlation between precipitation events and low potential evaporation. On rainy days the dynamic potential evaporation is generally lower, which decreases the amount of interception evaporation. Under climatological forcing the energy available is not reduced and thus more water evaporates from the interception store. The latter is sometimes taken into account in hydrological models by adding a potential evaporation reduction function dependent on the intensity of precipitation to correct the PET climatology. For example, the HBV model has this option (Schellekens et al., 2017).

The lower evaporation with dynamic PET forcing cascades through the different model storages, accumulating in a mostly wetter lower zone (LZ) storage under dynamic forcing. Finally, the lower evaporation results in higher discharge throughout the Rhine basin (see fig. S1-S6 in the supplemental information). Exceptions are the high Rhine during spring and to a lesser extent during autumn, and several areas during winter when there is very little effect overall. The wetter conditions also result in higher peak discharges. As these higher discharges are a result of the temporal dynamics of the potential evaporation input, it is expected to find a similar effect on forecasted discharges. As will be shown later (fig. 9), this is indeed the case.

## 4.3 Influence of PET forecast on streamflow forecast

The CRPSS for streamflow forecast is hardly influenced by potential evaporation forcing type. At first sight, the skill scores obtained with dynamic or climatological PET are identical. Small differences only become visible when taking a close up of the differences by substracting one from the other (fig. 8). However, the small difference in skill grows with lead-time. The influence of PET forcing type becomes more intuitive when looking at the relative mean error (RME). Visible is an increasing drift with lead-time between PET forcing types (fig. 9). Interestingly, this drift in RME is almost uniform over the whole distribution of predicted discharge. The drift is positive, which means that forecasted PET leads to slightly higher forecasted discharges, as expected based on the results of the influence of variable PET on the initial states.

Analyzed for each season separately, there is a little more to discover about the role of potential evaporation forecasts and the sensitivity of forecast skill to the meteorological forecast in general. The contribution of the meteorological forecast to streamflow forecast uncertainty is largest for summer, as shown by the largest decrease in CRPSS for the 10-day forecast in summer compared to the other seasons. The CRPSS especially 'dips' for the most extreme discharges, which is not as strong for spring and autumn, and especially compared to the flat response of the CRPSS for the highest 30% of discharges in winter.

In terms of the effect of potential evaporation climatology versus forecasted potential evaporation, the influence is largest (but still quite small) for summer and spring. This is tied to the potential evaporation being of larger magnitude; there is hardly a response for winter where there is lowest potential evaporation.



**Figure 7.** Seasonal Mean Difference in calculated actual evaporation (AET) for each season. Actual evaporation includes evaporation from interception.





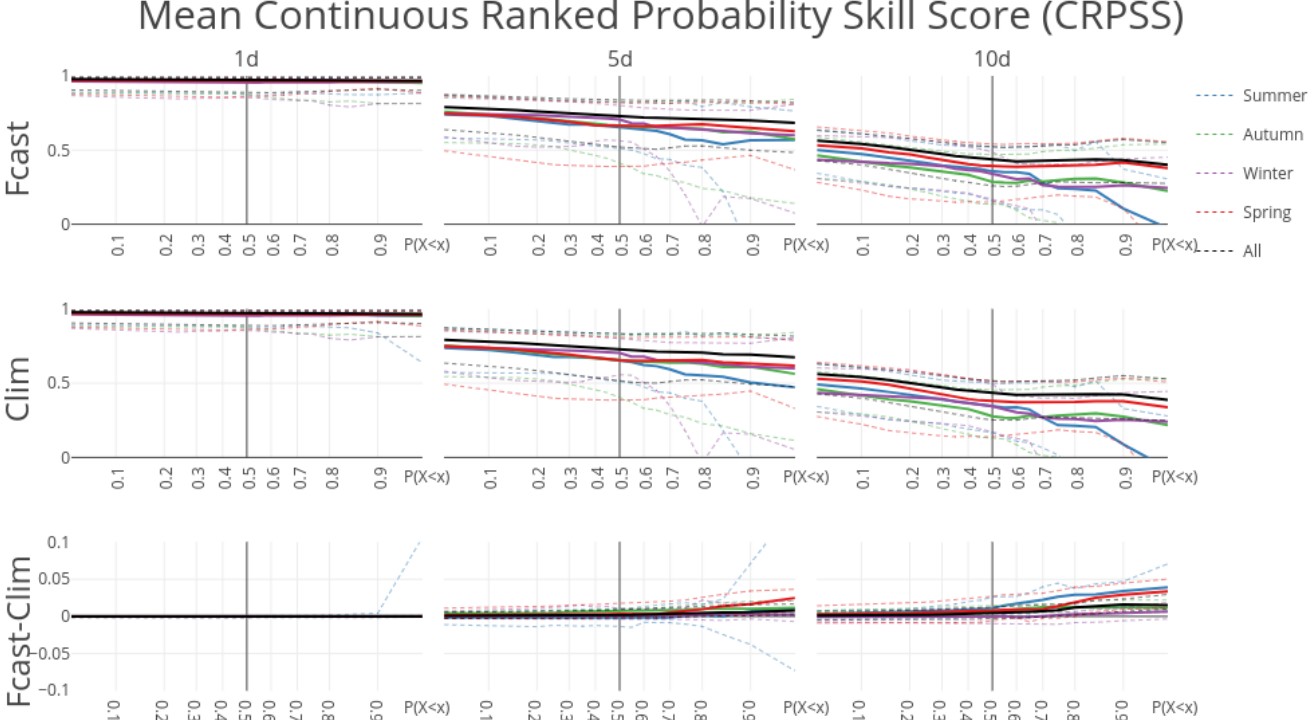

**Figure 8.** Continuous Ranked Probability Skill Score (CRPSS) for forecast runs (forecasted PET, climatological PET) and their difference benchmarked against model output for the 20 largest streams in the Rhine basin. CRPSS scores are aggregated into mean (solid), 10th and 90th percentile (dashed).

# 5 Conclusions

This paper presented a simple and straightforward investigation with an operational forecasting practice perspective. First, observation data was preprocessed for used in the gridded wflow_hbv model. Second, the wflow_hbv model was subjected to dynamical and climatological PET forcing. Three aspects were analyzed: 1) the skill in meteorological forecast, 2) the effect
5  of PET forcing on initial states and 3) the effect of PET forcing on forecast skill.

Nine to ten days is the upper limit on forecast lead time for daily precipitation for the ECMWF forecast in the Rhine basin, with only very little skill remaining compared to climatology. There is considerable skill in daily temperature, radiation and potential evaporation forecasts, also after ten days. Variable PET forcing resulted in lower evaporation and to wetter initial states and higher modeled discharges.

10  The main result of this study is that potential evaporation forecasts improved streamflow forecasts only slightly. This confirms earlier results that the influence of random errors on estimated streamflow was generally not measurable when comparing model runs directly, needing a 20% systematic bias in PET to influence model outcomes significantly (Parmele, 1972). Like-




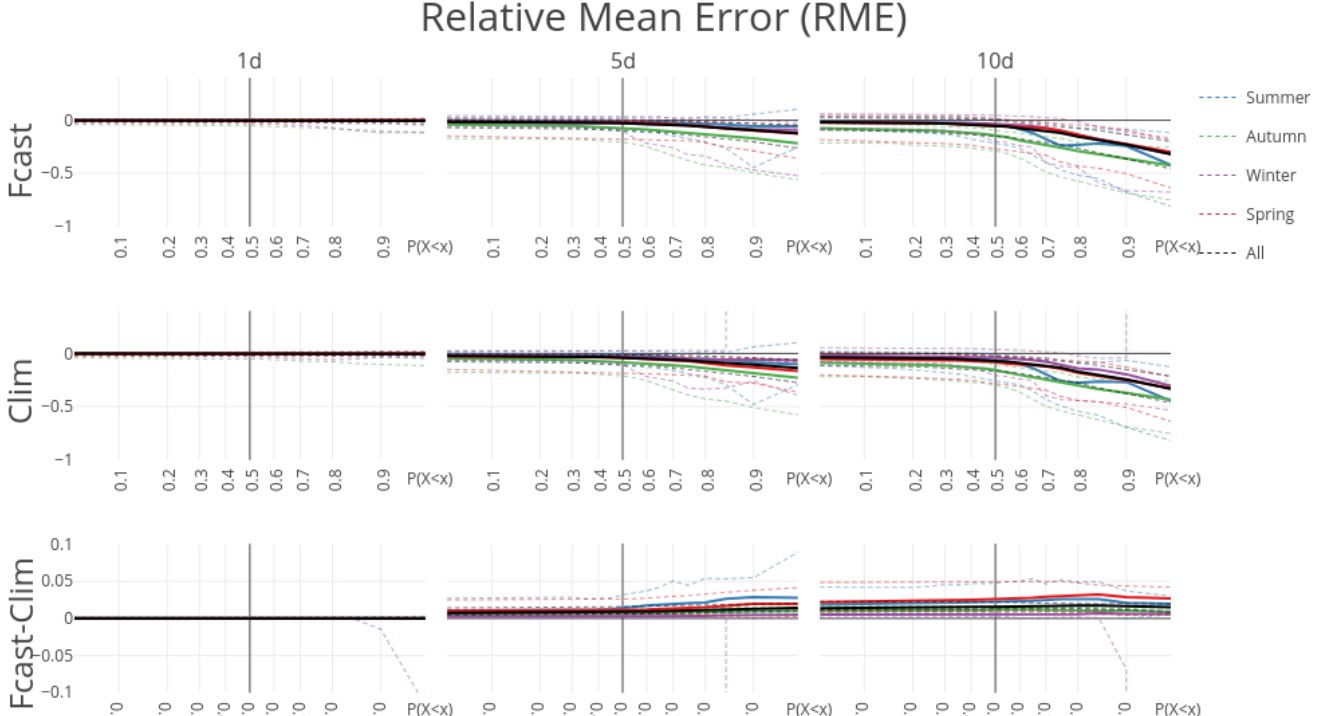

**Figure 9.** Relative Mean Error (RME) for forecast runs (forecasted PET, climatological PET) and their difference benchmarked against model output for the 20 largest streams in the Rhine basin. RME scores are aggregated into mean (solid), 10th and 90th percentile (dashed).

wise, Fowler (2002) concluded that climatological PET estimates produced a soil water regime very similar to that derived with actual daily PET values, including extreme periods, for a site in Auckland, New Zealand.

There is a wider discussion on evaporation modeling in hydrological models (Andréassian et al., 2004; Oudin et al., 2005a, b) to which the results here might add a new perspective: that of evaporation as a process relevant for medium term forecasts. This is directly also a limitation of this research; Only the influence on forecasts up to 10 days was investigated. The influence on seasonal forecasting might be more substantial, considering that the modeling of evaporation strongly influences the partitioning between runoff and evaporation in the longer term water balance (Bai et al., 2016).

Further limitations are that only one model was tested (wflow_hbv) and for one climate zone (moderate temperate). The model was calibrated originally on a different PET climatology than studied here and was not recalibrated. The latter is not seen as a limitation. Deliberately not recalibrating the model enabled to focus on the changes in modeled processes instead of comparably vague assessments based on model performance expressed in efficiencies, with the effects brought forward by the PET forcing somewhere hidden in the parameter space.

The idea to look at potential evaporation forecast was instigated as part of a program to improve forecasts of low flows. Indeed, it is a recurring hypothesis that potential evaporation forecasts should aid especially in making low flow predictions.





The uniform response of several skill scores for every percentile of observed discharge does not support this idea; there is no special gain for low flows. Instead from our model results it follows that the correct prediction of a drought is firstly dependent on a correct forecast of no-rain. Low flow recession is subsequently determined, in the absence of further feedback mechanisms, solely by the storage-discharge relationship of, in this case, the lower zone representing the saturated zone.

The follow-up question then is: Is this true in reality, or is this a model deficiency? Should we rethink hydrological modeling to incorporate more feedbacks on evaporation? Certainly there are models with more complex representation of evaporative processes. These are valid and important questions especially in the light of hydrologic response to change of climate drivers. However, from the results presented here, it should not be expected that a better understanding of evaporative processes and feedbacks will result directly in a significant increase in 10-day predictive skill for streamflow.

*Data availability.* Gridded precipitation, temperature, radiation and potential evaporation used in this study are available through the 4TU datacenter, see van Osnabrugge (2017, 2018).

*Competing interests.* The authors declare that they have no conflict of interest.

*Acknowledgements.* This work is partly supported by the IMPREX project funded by the European Commission under the Horizon 2020 framework program (grant 641811) and partly by the Dutch Ministry of Infrastructure and the Environment. Meteorological data for this
research have been gratefully received from the Deutscher Wetterdienst Climate Data Center; KNMI Data Centrum; Météo France; Federal Office of Meteorology and Climatology MeteoSwiss; Administration de la gestion de l'eau du Grand-Duché de Luxembourg; and Service Publique de Wallonie Département des Etudes et de l'Appui à la Gestion. Discharge data have been gratefully received from SCHAPI (Service Central d'Hydrométéorologie et d'Appui à la Prévision des Inondations) through 'Banque HYDRO'; Bundesamt für Umwelt BAFU; Bundesanstalt für Gewässerkunde (BfG); Administration de la gestion de l'eau du Grand-Duché de Luxembourg; Bavarian En-
vironment Agency, www. lfu.bayern.de; Landesanstalt für Umwelt, Messungen und Naturschutz Baden-Württemberg, LUBW; Landesamtes für Umwelt, Wasserwirtschaft und Gewerbeaufsicht Rheinland-Pfalz; Landesamt für Umwelt- und Arbeitsschutz Saarland; and Landesamt für Natur, Umwelt und Verbraucherschutz Nordrhein- Westfalen. We thank David Lavers from ECMWF for providing access to the reforecast dataset.



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
