# Peer review of "Contribution of Potential Evaporation Forecasts to 10-day streamflow forecast skill for the Rhine river"

_Hydrology and Earth System Sciences, 2018_

## Referee Comment (RC1) · Anonymous Referee #1 · 15 Nov 2018

In this paper, the authors explain the reason for their study by the fact that the influence of precipitation on forecasts has been studied most widely, while potential evapotranspiration (PET) have received little attention. Thus, they use a 20–year dataset of weather reforcasts and a daily hydrological model and to assess the importance of PET, they compare forecasts obtained with PET climatology (non-dated PET) with observation-based (dated) estimates of PET.

The authors modestly claim a "simple and straightforward investigation with an operational forecasting practice perspective".

The paper is perfectly written (I found only one typo p.6 l.10 "geostationairy"), and I

see nothing to add or change, except perhaps the reference to Makkink's paper in esperanto. In a century where young people often ignore this idealist-utopist linguistic movement, I think that Makkink's efforts could be rewarded by a citation !

The only critic that I could have made was that the conclusions are rather obvious... exactly the critic that I got for my 2004 paper on this topic... However I remember that I did not like the critic, so I withdraw mine. Moreover, what is obvious for older hydrologists is not obvious for everybody, and like André Gide wrote "everything has already been said, but since no one listens, one must always start again".

––––––––––––––––––––––––––––––

---

## Author Comment (AC1) · 16 Nov 2018

Thank you for the positive feedback.

We will add the reference to the Esperanto article. (Makkink, G.F.. (1957). Ekzameno de la formula de Penman. Neth. J. Agric. Sci. 5. 290-305.) It is interesting to realize that even though I never studied Esperanto any further than "Mi estas Bart" it is still readable for me. Indeed it is fitting to cite at least one Esperanto article in a study supported by the EU. (And it is also the most accurate reference according to (Mcmahon, Thomas & Malano, Hector & Schultz, B. (2015). Comment on the Reference to Makkink Potential Evaporation Equation. Journal of Irrigation and Drainage Engineering. 141.

[Figure]

02514001. 10.1061/(ASCE)IR.1943-4774.0000845. )

Of course the typo will also be corrected.

Kind regards,

Bart van Osnabrugge
* * *

---

## Referee Comment (RC2) · Anonymous Referee #2 · 23 Nov 2018

Review of "Contribution of Potential Evaporation Forecasts to 10-day streamflow forecast skill for the Rhine River"

This manuscript describes an investigation in the role of potential evapotranspiration (PET) forecasts in generating 10 ensemble streamflow forecasts. The authors show that there is skill in ECMWF PET forecasts, and that this influences the state variables of the hydrological model, but that there is only limited effect streamflow forecasts. This is an important topic that does not appear to have been evaluated from a forecasting perspective previously and therefore is material worthy of publication. Overall, I think the manuscript is well structured and presents the material in a logical manner. However, I believe there are a couple of technical issues that need to be carefully considered. The manuscript could also do with some copy editing.

In two places the authors (page14 line 13-15; page 19 paragraph 1) that authors make comment on the performance of forecasts for low observed values, on occasion contrasting the performance with forecasts for high observed values. The figures presented do not provide any information about the performance of forecasts for low observed values at all, only for all data (when $P(X<x)$ is near zero) and for increasingly high observed values (when $P(X<x)$ takes high values. To provide insight into the forecast performance of low values the figures need to be generated using ($P(X>x)$), rather than $P(X<x)$. Given that there are differences in the soil stores using the different forecasts PET forcing, then I would have expected there to be differences identified in streamflow forecasts for low flow conditions, and believe that some additional analysis that exclusively evaluates the low flow performance of forecasts needs to be undertaken and presented to support the statements that are made.

All forecast verifications presented in the manuscript are conditioned on observations. While this type of analysis is has been reported by several other authors, e.g. (Brown et al., 2012; Verkade et al., 2013) there are difficulties when interpreting this type of analysis that are very nicely described by (Lerch et al., 2017). When a forecast is issued, a forecast user only has knowledge of the forecast and not the observation. Therefore an analysis of forecast performance conditioned on observations cannot provide forecast users with an understanding of how well a given forecast may perform. A more robust approach, which can be directly interpreted by forecast users, is to condition the performance evaluation on the forecasts rather than the observations.

Editorial suggestions: Page 2 line 14: "monthly potential evaporation climatology forcing"

Page 3 line 12 "For this study the precipitation dataset is used which has been was derived by . . ."

Page 3 line 17 "correction for height elevation using"

Page 6 line 21 lambda is spelt out rather than using the symbol.

Page 8 line 9 question mark in reference

Page 9 line 8 "both scores have no fixed limit upper bound"

Figure 5 caption – remove "benchmarked against sample climatology" as these are raw CRPS scores.

There are many other examples where editing is required.

References:

Brown, J. D., Seo, D.-J., and Du, J.: Verification of Precipitation Forecasts from NCEP's Short-Range Ensemble Forecast (SREF) System with Reference to Ensemble Stream-flow Prediction Using Lumped Hydrologic Models, J. Hydrometeorol., 13, 808-836, 2012.

Lerch, S., Thorarinsdottir, T. L., Ravazzolo, F., and Gneiting, T.: Forecaster's Dilemma: Extreme Events and Forecast Evaluation, Statist. Sci., 32, 106-127, 2017.

Verkade, J. S., Brown, J. D., Reggiani, P., and Weerts, A. H.: Post-processing ECMWF precipitation and temperature ensemble reforecasts for operational hydrologic forecasting at various spatial scales, Journal of Hydrology, 501, 73-91, 2013.

––––––––––––––––––––––

---

## Author Comment (AC2) · 8 Jan 2019

Thank you for the review. The three main issues mentioned by this reviewer are: 1) figures generating P(X>x) rather than P(X<x), 2) Forecast verifications conditioned on forecasts can be a more robust approach, 3) copy editing. In the following we will address each point. At the end we address also the minor issues raised by the reviewer. The review is also added as pdf including the figures mentioned.

[Figure]

Creative Commons BY license logo

**1 P(X>x) for claims on low flows**

Although the comment about P(X<x) versus P(X>x) for evaluation of high and low flows respectively is correct in the sense that low P(X<x) evaluates for all data and not directly on low flows, there is still information about low flow forecasts to be found, namely in the shape of the curve.

We extracted information about low flows from the evaluation with P(X<x) by looking at the change in predictive skill over P(X<x). For p.14 l.13-15: "In terms of bias, low values are slightly overestimated while high values are slightly underestimated, making the radiation forecasts slightly less extreme than the observations (fig.6)" we see in fig. 6 (row 3, Rg) that the relative mean error increases with lower P(X<x). So how more low flow occurrences are added to the evaluated set, the more the relative mean error increases. This is only possible if low values are overestimated, which is what we claim. We understand that the explanation we give here was lacking so we will add this explanation.

Additionally, we performed the proposed analysis and plotted graphs for P(X>x) which showed that indeed low values are overestimated for Rg. (see attached Figure 1). However we think that adding the full explanation as written above is the best solution while adding the 'inverse' graph to the supplemental information including some explanatory text as not to disturb the flow of the article.

For p.19 l.1-3 "The uniform response of several skill scores for every percentile of observed discharge does not support this idea; there is no special gain for low flows" we would like to refer back to fig. 9 and apply a reasoning similar to the one above. First we note that the scale of the last row where the differences are plotted is very, very small. Even if 'diluted' by the other observations in the set, a significant change for low flow values should show in this figure. Instead, the more low flow values are added to the evaluation set, the smaller the RME difference becomes.

Also here we did the proposed analysis to confirm our statement, see attached Figure 2. Here we see our initial conclusions confirmed as the differences are negligible between PET forcings. Additionally, the analysis gives some new insight in the sensitivity of low flows to PET forcing. Looking at the 5d and 10d skill score, there is almost no loss of skill due to the combined forcing. In other words, the skill of an actual forecast is purely determined by the quality of the model and initial state. This can be readily explained by the fact that the lowest flows are caused by long periods of no rain. We will describe this in the result section.

Concerning the expectations that "Given that there are differences in the soil stores using the different forecasts PET forcing, then I would have expected there to be differences identified in streamflow forecasts for low flow conditions", we would like to respond that this indeed was our initial expectation and is indeed an expectation that is shared with many. It is one of the findings of this paper that this belief is untrue, at least for the conceptualization of the HBV model. This is because under dry conditions the HBV discharge is determined by the LowerZone storage and routing, without strong feedback mechanisms that would drain the LowerZone through evaporation. We are happy to have carried out the suggested analysis because this has become now much more clear in the results. Note that we did look at 10-day forecasts and that all forecasts are run from the same initial state created, so the difference in state is deliberately not taken into account to isolate the effect of the forecasted forcing.

**2   Verification conditioned on forecasts**

If we understand correctly, you mean that we should take samples based on the forcasted values, $P(F<f)$, instead of observations, $P(X<x)$, so that scores are calculated for the 10% (etc) highest (lowest) forecasts and not for the highest (lowest) 10% observations. This then will inform the forecaster about the forecast quality based on the
extremity of the forecast, not the unknown observation.

We studied the referenced paper with great interest. We see that such an analysis has merit, but do not think that it will add to the topic of this paper which is focused on the effect of evaporation forecasting on streamflow forecasting. In particular we do not think that our conclusions are susceptible to the danger of evaluating models on only a subset of the data because we did calculate our metrics over the whole range of $P(X<x)$, and after your first suggestion for $P(X>x)$, and are not tuning our model. We have however added a recommendation for this analysis of forecast skills for future studies to further the awareness of this issue, including a reference to the mentioned article(s).

**3   Copy editing**

We will thoroughly check the manuscript for copy editing errors to our best efforts. We thank the reviewer for already pointing out several cases that need our attention.

**Supplement:**

**Contribution of Potential Evaporation Forecasts to 10-day streamflow forecast skill for the Rhine river - Answer to reviewer 2**

Bart van Osnabrugge[1,2], Remko Uijlenhoet[2], and Albrecht Weerts[1,2]

[1]Deltares, Operational Water Management Department, Delft, The Netherlands
[2]Wageningen University, Hydrology and Quantitative Water Management Group, Wageningen, The Netherlands

**Correspondence:** Bart van Osnabrugge (Bart.vanOsnabrugge@deltares.nl)

Thank you for the review. The three main issues mentioned by this reviewer are: 1) figures generating P(X>x) rather than P(X<x), 2) Forecast verifications conditioned on forecasts can be a more robust approach, 3) copy editing. In the following we will address each point. At the end we address also the minor issues raised by the reviewer.

**1 P(X>x) for claims on low flows**

Although the comment about P(X<x) versus P(X>x) for evaluation of high and low flows respectively is correct in the sense that low P(X<x) evaluates for all data and not directly on low flows, there is still information about low flow forecasts to be found, namely in the shape of the curve.

We extracted information about low flows from the evaluation with P(X<x) by looking at the change in predictive skill over P(X<x). For p.14 l.13-15: "In terms of bias, low values are slightly overestimated while high values are slightly underestimated, making the radiation forecasts slightly less extreme than the observations (fig.6)" we see in fig. 6 (row 3, Rg) that the relative mean error increases with lower P(X<x). So how more low flow occurrences are added to the evaluated set, the more the relative mean error increases. This is only possible if low values are overestimated, which is what we claim. We understand that the explanation we give here was lacking so we will add this explanation.

Additionally, we performed the proposed analysis and plotted graphs for P(X>x) which showed that indeed low values are overestimated for Rg. (see attached Figure 1). However we think that adding the full explanation as written above is the best solution while adding the 'inverse' graph to the supplemental information including some explanatory text as not to disturb the flow of the article.

For p.19 l.1-3 "The uniform response of several skill scores for every percentile of observed discharge does not support this idea; there is no special gain for low flows" we would like to refer back to fig. 9 and apply a reasoning similar to the one above. First we note that the scale of the last row where the differences are plotted is very, very small. Even if 'diluted' by the other observations in the set, a significant change for low flow values should show in this figure. Instead, the more low flow values are added to the evaluation set, the smaller the RME difference becomes.

Also here we did the proposed analysis to confirm our statement, see attached Figure 2. Here we see our initial conclusions confirmed as the differences are negligible between PET forcings. Additionally, the analysis gives some new insight in the

sensitivity of low flows to PET forcing. Looking at the 5d and 10d skill score, there is almost no loss of skill due to the combined forcing. In other words, the skill of an actual forecast is purely determined by the quality of the model and initial state. This can be readily explained by the fact that the lowest flows are caused by long periods of no rain. We will describe this in the result section.

5      Concerning the expectations that "Given that there are differences in the soil stores using the different forecasts PET forcing, then I would have expected there to be differences identified in streamflow forecasts for low flow conditions", we would like to respond that this indeed was our initial expectation and is indeed an expectation that is shared with many. It is one of the findings of this paper that this belief is untrue, at least for the conceptualization of the HBV model. This is because under dry conditions the HBV discharge is determined by the LowerZone storage and routing, without strong feedback mechanisms that 10   would drain the LowerZone through evaporation. We are happy to have carried out the suggested analysis because this has become now much more clear in the results. Note that we did look at 10-day forecasts and that all forecasts are run from the same initial state created, so the difference in state is deliberately not taken into account to isolate the effect of the forecasted forcing.

**2   Verification conditioned on forecasts**

15   If we understand correctly, you mean that we should take samples based on the forcasted values, P(F<f), instead of observations, P(X<x), so that scores are calculated for the 10% (etc) highest (lowest) forecasts and not for the highest (lowest) 10% observations. This then will inform the forecaster about the forecast quality based on the extremity of the forecast, not the unknown observation.

     We studied the referenced paper with great interest. We see that such an analysis has merit, but do not think that it will add 20   to the topic of this paper which is focused on the effect of evaporation forecasting on streamflow forecasting. In particular we do not think that our conclusions are susceptible to the danger of evaluating models on only a subset of the data because we did calculate our metrics over the whole range of P(X<x), and after your first suggestion for P(X>x), and are not tuning our model. We have however added a recommendation for this analysis of forecast skills for future studies to further the awareness of this issue, including a reference to the mentioned article(s).

**25   3   Copy editing**

We will thoroughly check the manuscript for copy editing errors to our best efforts. We thank the reviewer for already pointing out several cases that need our attention.

[Figure]

**Figure 1.** Relative Mean Error (RME) for the four forcing variables benchmarked against sample climatology for the 148 HBV subbasins for the whole year. RME is aggregated into mean (solid), 10th and 90th percentile (dashed). Note that radiation (Rg third row) is indeed overestimated for low extremes as presented in the main text. Additionally, the asymptotic behaviour of the RME of precipitation (P, first row) is caused by the large number of zero or close to zero events so that the relative error grows without bounds. In the inverse figure (for P(X<x), Fig. 6) those zero values were automatically excluded. For temperature (T, second row) the RME is unstable for values around zero, but since actual exactly zero temperatures are rare, this remains within bounds albeit with a jump from positive to negative due to sign differences between observed and forecasted values.

[Figure]

**Figure 2.** Continuous Ranked Probability Skill Score (CRPSS) for forecast runs (forecasted PET, climatological PET) and their difference benchmarked against model output for the 20 largest streams in the Rhine basin. CRPSS scores are aggregated into mean (solid), 10th and 90th percentile (dashed). Note that this is the inverse graph of Figure 8 (for P(X<x)) in the main text.